# Polymerizable rotaxane hydrogels for three-dimensional printing fabrication of wearable sensors

Xueru Xiong[1,2,7], Yunhua Chen[1,2,3,7], Zhenxing Wang[1,2], Huan Liu[2], Mengqi Le[2], Caihong Lin[1,2], Gang Wu[1,2], Lin Wang[1,2,3] ✉, Xuetao Shi[1,2,4] ✉, Yong-Guang Jia[1,2,5] ✉ & Yanli Zhao[6] ✉

While hydrogels enable a variety of applications in wearable sensors and electronic skins, they are susceptible to fatigue fracture during cyclic deformations owing to their inefficient fatigue resistance. Herein, acrylated β-cyclodextrin with bile acid is self-assembled into a polymerizable pseudorotaxane via precise host-guest recognition, which is photopolymerized with acrylamide to obtain conductive polymerizable rotaxane hydrogels (PR-Gel). The topological networks of PR-Gel enable all desirable properties in this system due to the large conformational freedom of the mobile junctions, including the excellent stretchability along with superior fatigue resistance. PR-Gel based strain sensor can sensitively detect and distinguish large body motions and subtle muscle movements. The three-dimensional printing fabricated sensors of PR-Gel exhibit high resolution and altitude complexity, and real-time human electrocardiogram signals are detected with high repeating stability. PR-Gel can self-heal in air, and has highly repeatable adhesion to human skin, demonstrating its great potential in wearable sensors.

The unique properties of hydrogels, such as superior softness, wetness, responsiveness and biocompatibility, have enabled a variety of applications in wearable sensors[1–3], implantable bioelectronics[4–6], and electronic skins[7–9]. Ideally, the essential requirement for hydrogel sensors is that the hydrogel can undergo deformation, such as stretching or twisting under repeated stress[10,11]. Conventional hydrogels with chemical crosslinking tend to be soft and brittle due to the inhomogeneous gel networks and the lack of effective energy dissipation mechanisms[12,13], which easily and irreversibly damage the hydrogel networks. Increasing efforts have been devoted to the development of highly stretchable hydrogels, and the tensile properties of hydrogels have made substantial progress in the applications of flexible sensors. However, even though their toughness could be engineered as high as that of natural rubbers by introducing sacrificial noncovalent bonds, such as double-network hydrogels[14,15], and nanocomposite hydrogels[16,17], most of these synthetic hydrogels are susceptible to fatigue fracture during their repeated stretching cycles owing to inefficient fatigue resistance[18,19]. Therefore, the design of hydrogels with excellent mechanical properties (high flexibility, resilience, etc.) and highly stable performance during the practical long-term applications of wearable sensors remains a tremendous challenge.

[1]School of Materials Science and Engineering, South China University of Technology, Guangzhou 510641, China. [2]National Engineering Research Center for Tissue Restoration and Reconstruction, South China University of Technology, Guangzhou 510006, China. [3]Key Laboratory of Biomedical Engineering of Guangdong Province, South China University of Technology, Guangzhou 510006, China. [4]Key Laboratory of Biomedical Materials and Engineering of the Ministry of Education, South China University of Technology, Guangzhou 510006, China. [5]Innovation Center for Tissue Restoration and Reconstruction, South China University of Technology, Guangzhou 510006, China. [6]School of Chemistry, Chemical Engineering and Biotechnology, Nanyang Technological University, 21 Nanyang Link, Singapore 637371, Singapore. [7]These authors contributed equally: X. Xiong, Y. Chen. ✉e-mail: wanglin3@scut.edu.cn; shxt@scut.edu.cn; ygjia@scut.edu.cn; zhaoyanli@ntu.edu.sg

The slide ring (SR) hydrogels with topological networks consisting of mechanical interlocked units (Fig. 1a) were initially developed by Ito and coworkers using the polyrotaxane of poly(ethylene glycol) (PEG) and α-cyclodextrin (α-CD)[20], in which the pulley effect created by the mobile junctions could effectively disperse the stress, resulting in high toughness and excellent fracture resistance. For example, the toughness of SR hydrogels was one order of magnitude higher than that of PEG covalently crosslinked hydrogels[21]. The SR topological supramolecular network was also introduced into the elastomers to enhance the toughness and fracture resistance. For instance, Gao et al. developed SR crosslinked poly(acrylic acid) as a self-adaptive interfacial layer for lithium anodes, which exhibited the remarkable anti-fatigue property and enabled it to respond rapidly to pressure and strain[22]. Recently, Bao and coworkers have achieved a conducting polymer with simultaneously high conductivity and stretchability in bioelectronic applications by introducing a rational SR network[23]. Therefore, SR topological supramolecular networks exhibit outstanding advantages in the mechanical properties of elastic materials compared with their conventional counterparts crosslinked with chemical networks. Despite these recent developments, designing and fabricating SR hydrogels with excellent fatigue resistance for hydrogel sensors, such as strain sensors in electronic skin, remains largely unexplored[24–27]. To date, detailed studies on SR topological supramolecular networks are mostly limited to the polyrotaxane of PEG and α-CD. Meanwhile, the synthetic protocol of polyrotaxane with low and precise host coverage is not only complicated but also inaccessible[21,28–31].

Bile acids (BAs), derived from cholesterol in mammals, such as cholic acid (CA) and lithocholic acid (LCA), are a group of previously unexplored and underrated guests for the preparation of topological networks in hydrogels[32–34]. Herein, a kind of polymerizable pseudorotaxane crosslinker of acrylated β-cyclodextrin (β-CD) with a BA derivative is assembled by precise host-guest recognition (Fig. 1b) and then photocured with acrylamide (Am) in a binary solvent system of ethylene glycol (EG)/water with the presence of choline chloride (ChCl) to construct conductive hydrogels (Fig. 1c), which are denoted as PR-Gel due to the topological networks of polymerizable rotaxane (PR). The polymerizable crosslinkers generate a great number of mobile junctions in PR-Gel similar as to those of SR hydrogels. The mobile junctions between β-CD and BA units simultaneously endow PR-Gel with important advancements in the applications of strain sensors. For example, PR-Gel can be stretched up to 830% and no significant hysteresis is observed even after 500 cycles at a strain of 300%, confirming its excellent stretchability along with superior fatigue resistance. PR-Gel based strain sensor with a gauge factor of 8.53 can sensitively detect and distinguish large body-motions and subtle muscle movements. The marriage of three-dimensional (3D) printing technology with PR-Gel realized personalized and rapid preparation of flexible strain sensors with complex geometry, which could detect human electrocardiogram signals in real-time with high repeating stability. In addition, the presence of ChCl and EG also provide the PR-Gel with moisture-preserving and antifreezing advantages. Moreover, the resulting PR-Gel exhibits suitable adhesion to human skin and self-

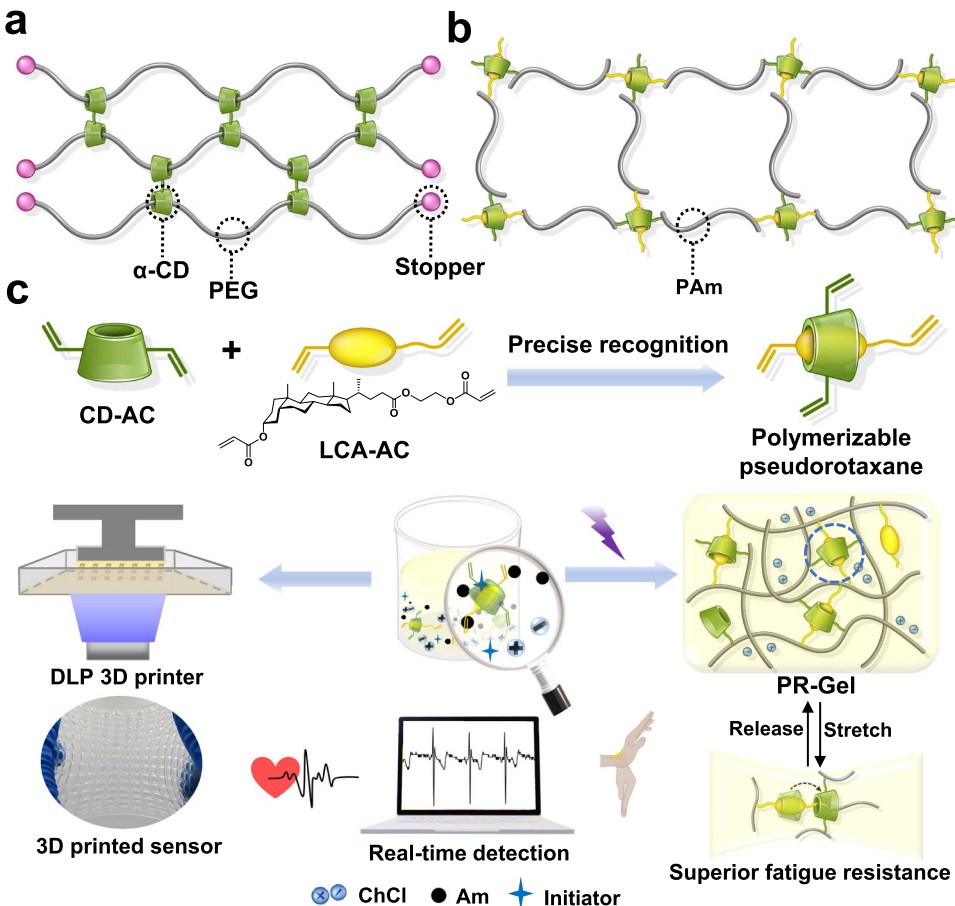

**Fig. 1 | Schemtic presentation of the design and applications of PR-Gel.** **a** Illustration of the topological networks constructed by the polyrotaxane of PEG with α-CD in SR hydrogels; **b** illustration of the topological networks of PR-Gel constructed by the polymerizable pseudorotaxane crosslinkers of BA with β-CD derivatives; **c** design and assembly of polymerizable pseudorotaxane crosslinker and further applications of superior fatigue-resistant hydrogels and 3D printing fabrication of wearable flexible sensors. DLP 3D printer and PAm refer to the digital light processing (DLP) based 3D printer and polyacrylamide (PAm), respectively. It should be noted that CD-AC is a mixture of acrylate modified β-CD and the acrylate units on each CD-AC were averaged to be ca. 1.9.

healing ability in air. This work provides a system of topological hydrogels without the supramolecular architecture of conventional polyrotaxane, but with excellent comprehensive performance, including extraordinary fatigue resistance, demonstrating a new paradigm for the design of 3D printable flexible devices.

## Results

### Assembly of the crosslinker and photocuring of PR-Gel

The pseudorotaxane crosslinker was assembled via precise host-guest recognition of β-CD with an LCA derivative as shown in Fig. 1c. The host molecule acrylate modified β-CD (CD-AC) and guest molecule ethylene glycol lithocholate derived diacrylate (LCA-AC) were both modified with acryloyl chloride (Supplementary Figs. 1–10), enabling the pseudorotaxane crosslinker photocurable. The $^1$H NMR spectrum could provide insights into the recognition of host and guest in aqueous solution[35,36], but the extremely low solubility of LCA-AC in water limited the ulitilization of $^1$H NMR characterization. Powder X-ray diffraction (XRD) measurements (Fig. 2a) were used to investigate the solid-state structures of pseudorotaxane crosslinker, where the diffraction peaks (2θ) of LCA-AC are observed at 13.9 and 15.4, and two broad peaks (2θ) at 12.5 and 18.4 are assigned to the characteristic diffraction peaks of CD-AC. All the diffraction peaks of LCA-AC completely disappear in pseudorotaxane. In contrast, the physical mixture of LCA-AC and CD-AC exhibits both characteristic diffraction peaks of LCA-AC and CD-AC. This indicates that CD-AC exclusively recognizes the LCA moiety of LCA-AC. The association constant $K_a$ of pseudorotaxane was further determined by isothermal titration calorimetry (ITC). The $K_a$ value for the complex (pseudorotaxane) of LCA with CD-AC is $2.36 \pm 0.18 \times 10^4$ M$^{-1}$ in the binary solvent system of EG/water (Fig. 2b), a value close to that of the LCA based polymer with β-CD in water (ca. $2.04 \pm 0.1 \times 10^4$ M$^{-1}$)[36].

To demonstrate the host-guest recognition of CD-AC and LCA-AC in the $^1$H NMR spectrum, the water-soluble counterpart of LCA-AC, LCA-AC-PEG with a PEG 2000 segment, was synthesized as shown in Supplementary Figs. 11–13. Obvious changes in the chemical shift of the methyl protons (peaks 18, 19, and 21 in Supplementary Fig. 14) in LCA units were observed in the $^1$H NMR spectrum of LCA-AC-PEG upon mixing with CD-AC, in which the proton signals of the three methyl groups on LCA units all shifted from peaks 18, 19 and 21 to peaks 18′, 19′, and 21′, respectively. Such $^1$H NMR spectral changes were further investigated by nuclear Overhauser enhancement spectroscopy (NOESY). The NOE correlation of protons from peaks 18′, 19′, and 21′ with the interior H3/H5 of the β-CD moiety on CD-AC (blue rectangles in Fig. 2c) confirmed the formation of the polymerizable pseudorotaxane crosslinker via host-guest recognition between the β-CD and LCA units.

The powder XRD spectra of the complex of LCA-AC-PEG with β-CD further confirmed that β-CD slid over the PEG segments and exclusively recognized LCA units, in which the characteristic diffraction peaks (2θ) of β-CD were observed at 8.9 and 12.4, and the peaks at 18.9 and 23.1 were assigned to the characteristic diffraction peaks of PEG segments. The diffraction peaks of β-CD completely disappeared in the complex and only the diffraction peaks of PEG remained (Supplementary Fig. 15). The association constant $K_a$ of the complex of LCA-AC-PEG with CD-AC is $6.09 \pm 0.6 \times 10^3$ M$^{-1}$ in water based on the results of ITC (Supplementary Fig. 16), which is slightly lower than that in the host-guest complex system between the LCA based polymer and β-CD[36].

The preparation process of PR-Gel and further applications in wearable flexible sensors are described in Fig. 1c. The pseudorotaxane crosslinker of CD-AC and LCA-AC was photocured with Am in the presence of conductive ChCl and in a binary solvent system of EG/water to obtain the conductive PR-Gel, in which the mobile junctions were introduced into the topological networks of hydrogels. For comparison, hydrogels crosslinked with N,N′-

methylenebisacrylamide (MBA) and CD-AC were prepared and denoted as MBA-Gel and CD-Gel (Supplementary Table 1), respectively, to reveal the contribution of PR topological networks to the properties of PR-Gel.

### Mechanical properties and fatigue resistance

The mechanical properties of the PR-Gel were dramatically improved with increasing concentrations of Am, as shown in Fig. 2d. For example, the fracture stress increased from 6.8 to 78.1 kPa as the concentration of Am increased from 20 w/v% to 40 w/v%, while the work of rupture also increased from 29.9 to 270.7 kJ/m$^3$. A higher concentration of Am, such as 50 w/v%, could lead to the precipitation of LCA-AC from the precursor solution. Therefore, the concentration of Am in the PR-Gel was set to 40 w/v% for the subsequent studies, in which 0.5 mol% of pseudorotaxane crosslinker was used to photopolymerize with Am, and the PR-Gel could be stretched up to 830%. Figure 2e confirmed that PR-Gel exhibited a higher tensile strength and elongation than MBA-Gel and CD-Gel at the same concentration of crosslinkers. Scanning electron microscope (SEM) images of lyophilized PR-Gel showed obviously different morphology features, compared with that of MBA-Gel and CD-Gel (Supplementary Fig. 17). There was an inverse dependence between the stiffness and toughness of MBA-Gel with a fixed crosslinking network[37,38], but PR-Gel exhibited an exceptional fracture behavior that contradicted this trade-off (Supplementary Fig. 18). This discrepancy is clearly attributed to the mobile junctions in the topological networks. The stress–strain curves of CD-Gel also differed from those of MBA-Gel, in which CD-Gel exhibited a higher elongation at break, but lower fracture stress. The freely mobile junctions assembled by β-CD and LCA units drastically enhanced the mechanical properties of polymeric materials, including both elongation and fracture stress.

The binary solvent system endows the PR-Gel with a great capability to tolerate harsh environmental conditions, showing high flexibility and stretchability down to −20 °C and up to 60 °C (Fig. 2f). Even after storage for 10 h at −20 °C, the stretchability of PR-Gel was only slightly reduced from 830 to 650% compared to that at 25 °C, indicating the outstanding anti-freezing performance. Similarly, the stretchability of PR-Gel remained almost unchanged at 790% after storage for 10 h at 60 °C. The excellent stretchability of PR-Gel over wide temperature ranges provided great convenience for the applications of flexible sensors under extreme conditions. Compression tests (Fig. 2g–i) further demonstrated the flexibility and toughness of PR-Gel. The cylindrical sample of PR-Gel was compressed at a strain of 95%, and then quickly returned to its original shape without any damage after the removal of external force, indicating that the PR topological networks significantly improved the compression resistance of the PR-Gel and exhibited exceptional shape recovery performance (Supplementary Movie 1).

Figure 2j schematically illustrates a comparison between the chemically crosslinked MBA-Gel and PR-Gel under tensile deformation. The tensile stress is concentrated in the short chains of MBA-Gel. In contrast, the mobile junctions in PR-Gel networks can disperse the stress in the polymer chains automatically during tensile deformation, resulting in a higher fracture resistance. To confirm the mobility of rotaxanes between β-CD and BA units in the deformation of PR-Gel, small angle X-ray scattering (SAXS) patterns of the stretched hydrogels are shown in Fig. 2k. As clearly seen, the SAXS pattern of the PR-Gel before elongation is isotropic. At strains of 100, 300, and 500%, they show no anisotropy in the binary solvent system of EG/water, indicating that the PR-Gels are highly homogeneous without higher order structure. These results were attributed to the fact that the polymer chains could pass through mobile junctions between β-CD and BA units to maximize the entropy, maintaining an isotropic structure even under a large elongation[39].

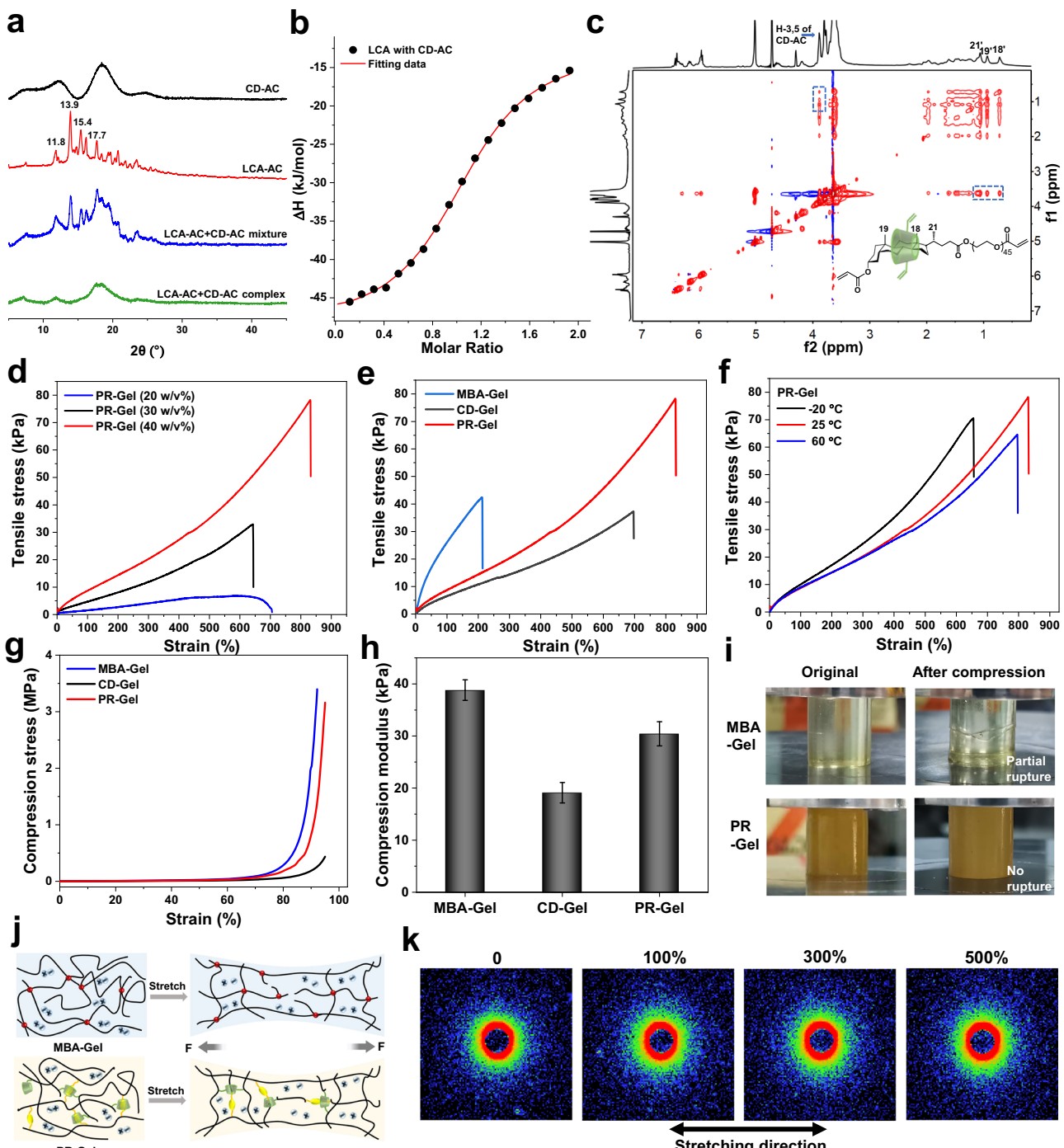

**Fig. 2 | Characterization of the key systems. a** Powder XRD patterns of CD-AC, LCA-AC, and a mixture of CD-AC with LCA-AC (1/1 mol.), and CD-AC/LCA-AC host-guest complex (1/1 mol.); **b** apparent reaction heat in calorimetric titrations of CD-AC solution (5.0 mM in EG/water) injecting into LCA-AC solution (0.5 mM in EG/water) at 25 °C and their typical ITC fitting curves; **c** 2D NOESY $^1$H NMR spectrum of the host-guest complex of CD-AC with LCA-AC-PEG in D$_2$O (1/1 mol., 25 °C); **d** tensile stress–strain curves of PR-Gel with various concentrations of Am (20 w/v%, 30 w/v% and 40 w/v%) at 25 °C; **e** tensile stress–strain curves of MBA-Gel, CD-Gel and PR-Gel with a concentration of Am 40 w/v% at 25 °C; **f** tensile stress–strain curves of PR-Gel with a concentration of Am 40 w/v% after storing for 10 h at −20, 25 and 60 °C, respectively; **g** compression curves and **h** compression modulus of MBA-Gel, CD-Gel and PR-Gel with a concentration of Am 40 w/v% at 25 °C; **i** photographs of MBA-Gel and PR-Gel with a concentration of Am 40 w/v% before and after compression at 25 °C; **j** schematic illustration of the crosslinked network structure of the MBA-Gel and PR-Gel; **k** SAXS patterns of the stretched PR-Gel. The sample is elongated in the horizontal direction in the images and the stretching ratios, include 0, 100, 300, and 500%.

In addition to flexibility and stretchability, antifatigue is also crucial to flexible sensors. The ideal polymer networks have equal length and uniformly distributed chains, which reveals a high resilience without significant energy dissipation during deformation. PR-Gel with mobile junctions can slide freely along the chain axis under external forces due to the permanent and slidable nature of the mobile junctions, balancing the stresses concentrated in the shorter chains, thereby causing the reconfigured gel networks to approximate the ideal polymer networks[40–42]. Figure 3a shows the loading–unloading cycle tests of the PR-Gel under various strains. Compared with MBA-

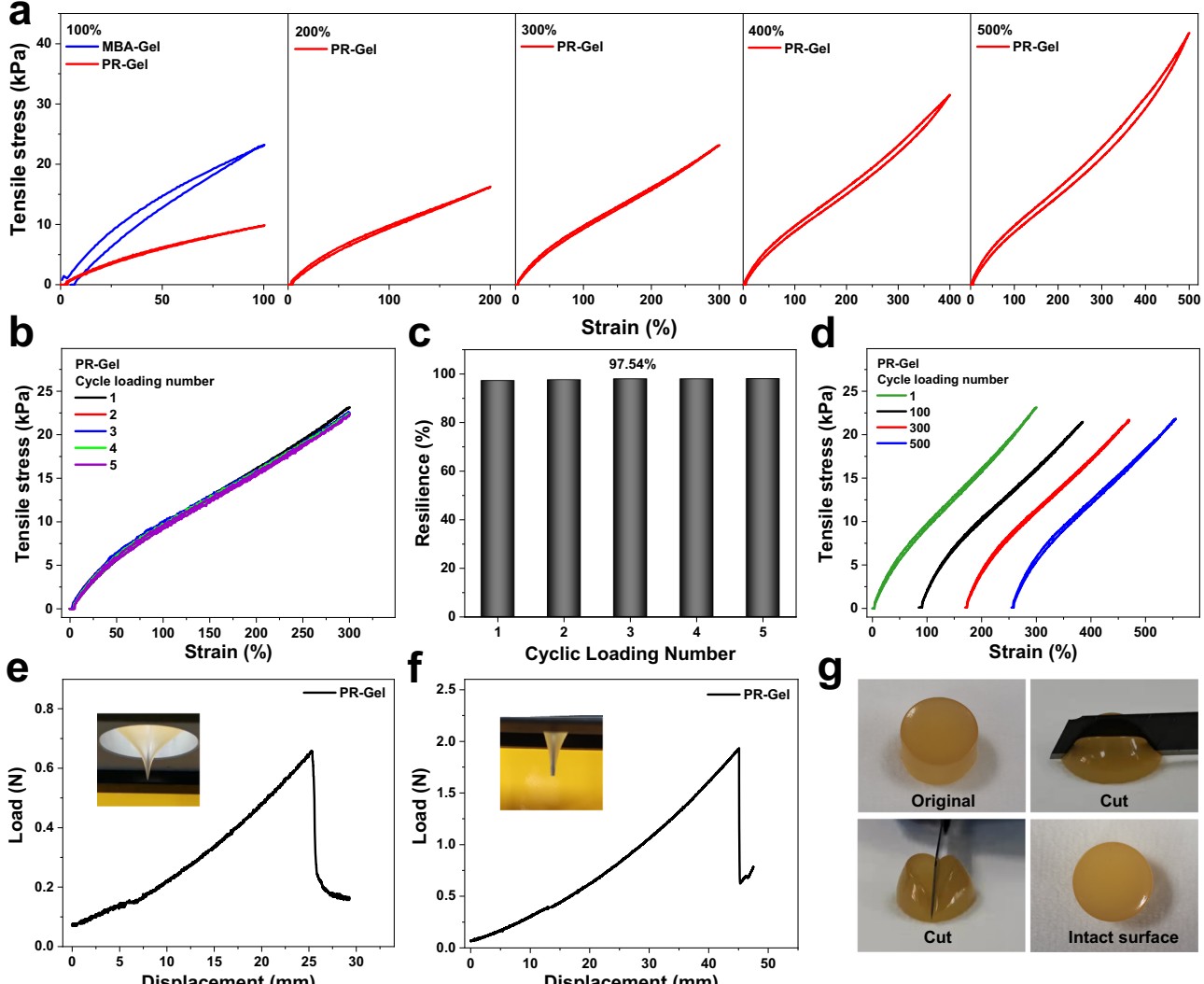

**Fig. 3 | Tensile studies. a** Tensile loading–unloading cycles of PR-Gel and MBA-Gel with a concentration of Am 40 w/v% at different strains and at 25 °C; **b** successive cyclic tensile tests and **c** resilience of PR-Gel (40 w/v%) for 5 cycles at 300% strain and at 25 °C; **d** successive cycle stress–strain curves of PR-Gel (40 w/v%) at 300% strain with the various numbers of cycles at 25 °C and the curves are horizontally offset for clarity; **e**, **f** puncture curves of PR-Gel (40 w/v%) under the metal needles with the outer diameters of 1 and 4 mm, respectively, at 25 °C; **g** photographs of PR-Gel (40 w/v%) cutting with a sharp knife at 25 °C.

Gel, the loading–unloading curve of PR-Gel at 100% strain exhibited no hysteresis. It still demonstrated a purely elastic mechanical response at a large deformation of 300%. For strains of 400 and 500%, minuscule hysteresis loops were observed during the loading–unloading cycle. From the results of five uninterrupted cyclic tensile experiments under a strain of 300% (Fig. 3b), it was evident that the loading–unloading curves of the first five cycles approximately overlapped, corresponding to dissipation energies between 0.82 and 1.11 kJ/m³ (Supplementary Fig. 19) and above 97% resilience (Fig. 3c), indicating that the PR-Gel had outstanding elastic and self-recovery properties under large deformation. To further test the fatigue resistance of the PR-Gel, no significant hysteresis was observed even after 500 cycles at 300% strain, as shown in Fig. 3d, indicating that the PR-Gel had extraordinary cycling stability and fatigue resistance.

Puncture experiments were performed with different sizes of metal needles on PR-Gel films as shown in Fig. 3e, f. With increasing loading stress, obvious umbrella-like puncture deformation was observed (Supplementary Movie 2). The puncture displacement amounted to 25 mm even with a slim needle of 1 mm diameter, indicating that the film had good puncture resistance for avoiding breakage by sharp objects in practical applications. The structure remained

intact without leaving any scars when it was strongly cut with a sharp blade (Fig. 3g). These phenomena were unobservable in the chemically crosslinked MBA-Gel. Therefore, the PR topological network is the key to these anomalous mechanical behaviors due to their mobile mechanical interlocking and strong interactions with polymer segments.

## Self-adhesion and self-healing

PR-Gel self-adhered firmly to the skin as shown in Fig. 4a, and more importantly, there was no residue or allergy on the adhered wrist. The adhesion of PR-Gel with nitrile gloves, glass plates, silicone rubber, aluminum and pigskin was quantified by lap-shear tests (Fig. 4b). Although the adhesion strength between PR-Gel and the substrates ranged from 4.5 to 6.0 kPa, and was lower than that of certain strongly adhesive hydrogels[43,44], this moderate self-adhesion allowed tight contact between PR-Gel and human skin as shown in Fig. 4a, ensuring stable signal transmission and repeatable adhesion for applications in wearable sensors.

Self-healing properties could expand the lifespan of wearable flexible sensors, which directly affects the reliability of the sensors in practical applications. Certain acrylates on LCA-AC may not

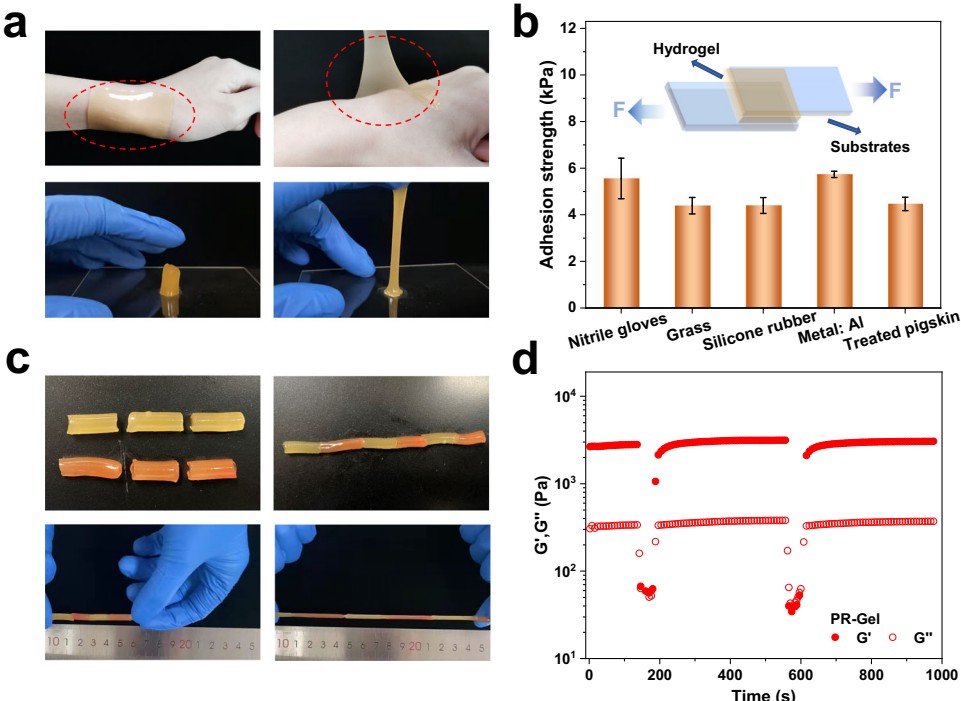

**Fig. 4 | Shear adhesive strength and self-healing. a** Images of PR-Gel (40 w/v%) adhered to wrist joints and glass surface; **b** shear adhesive strength of PR-Gel (40 w/v%) on substrates of nitrile glove, glass plate, silicone rubber, aluminum and pigskin. Inset: Schematic illustration of the lap-shear test at 25 °C; **c** self-healing behavior of the PR-Gel (40 w/v%) at 25 °C (colored with rhodamine B for visualization); **d** storage modulus ($G'$) and loss modulus ($G''$) of the PR-Gel (40 w/v%) at alternate strains between small (1%) and large strain (500%) at 25 °C.

photopolymerize, and then form host-guest dynamic complexation with CD moieties, which plays an important role in the self-healing behavior of PR-Gel. The hydrogen bonds between polyacrylamide chains may also contribute to the dynamic network rebuilding of PR-Gel. Meanwhile, the polymer chains of PR-Gel could pass through the crosslinks to maintain an isotropic structure as shown in Fig. 2k, which enhances the spatial network fluidity of hydrogel and is conducive to the improvement of self-healing performance[45].

Two cylindrical samples of PR-Gel were cut and then reassembled along the cut to qualitatively demonstrate the self-healing properties. For better visualization, one of the samples was stained red-orange as shown in Fig. 4c, and the healed gels could withstand tensile deformation without fracture at ambient temperature. The self-healing performance of PR-Gel was further evaluated, and structural damage and recovery tests of PR-Gel were executed by applying continuous strain sweeps with the alternating low (1%) and high (500%) oscillatory excitations. Figure 4d shows the storage modulus ($G'$) and loss modulus ($G''$) of PR-Gel under continuous strain sweeps, demonstrating that the PR-Gel had the rapid self-healing properties after damage. This process could be repeated several times, indicating that the recovery of the PR-Gel network was reproducible. In contrast, MBA-Gel showed the typical brittle behavior of chemically crosslinked hydrogels (Supplementary Fig. 20).

### Sensitivity, stability and detection of human motion

To visualize the conductivity of PR-Gel, a circuit with a light-emitting diode (LED) was designed as shown in Fig. 5a. The stretching of the PR-Gel caused resistance variation that altered the brightness of the LED indicator. After cutting, PR-Gel could self-heal to reconnect the circuit and still had mechanical strength to withstand stretching, demonstrating that PR-Gel had a rapidly self-healable conductivity. PR-Gel achieved a high ionic conductivity of 0.93 S/m at room temperature and maintained a level of 0.54 S/m even at −20 °C due to the excellent anti-freezing performance (Supplementary Fig. 21)[46,47]. To evaluate the

conductive sensitivity of PR-Gel, the gauge factor (GF) was calculated from the linear fitting curve of relative resistance change ($\Delta R/R_0$) versus different strains. Figure 5b indicates the dependence of GF on the various strains, in which the GF was estimated to be ca. 3.91 in the 200% strain range, increased to 6.43 at 200–400% strain, and further grew to 8.53 at 400–500% strain, indicating the high sensitivity and wide response range of PR-Gel. Only minor hysteresis was observed in the electrical signal during the loading–unloading cycle even at a large strain of 500% (Fig. 5c), which was consistent with the results of superior fatigue resistance. Moreover, compared with the original state, the curves still showed a large overlap after one month of indoor placement (Fig. 5d) due to the favorable water retention capacity of PR-Gel. The GF values under different strains were comparable to those in the original state. For example, the GF value showed a negligible decrease of ca. 1.1% under a strain of 200–400% (Fig. 5e), indicating the long-term stability of the PR-Gel. To further evaluate the stability and durability, PR-Gel was subjected to 100 successive tensile cycles at a strain of 300%. Highly reproducible resistance signals were obtained in each cycle, as shown in Fig. 5f, in which not only was the resistance signal highly repeatable, but also the baseline was extremely stable. Such performance may be attributed to the topological networks of PR-Gel simultaneously rendering the stable ion transportation and excellent anti-fatigue properties of PR-Gel.

Based on their excellent comprehensive performance, such as extraordinary fatigue resistance, high strain sensitivity, remarkable electrical stability, and fast self-adhesion and self-healing capability, PR-Gel-based wearable strain sensors would be suitable for the accurate real-time detection of human motions. Figure 6a shows the rate of resistance change of the PR-Gel-based strain sensor under different finger bending conditions. When the finger was bent at different angles, the strain sensor was stretched, thus increasing resistance. Holding the finger at a certain angle, the resistance value of the strain sensor was constant due to its good electrical stability, and the resistance change regained the original value immediately when the finger

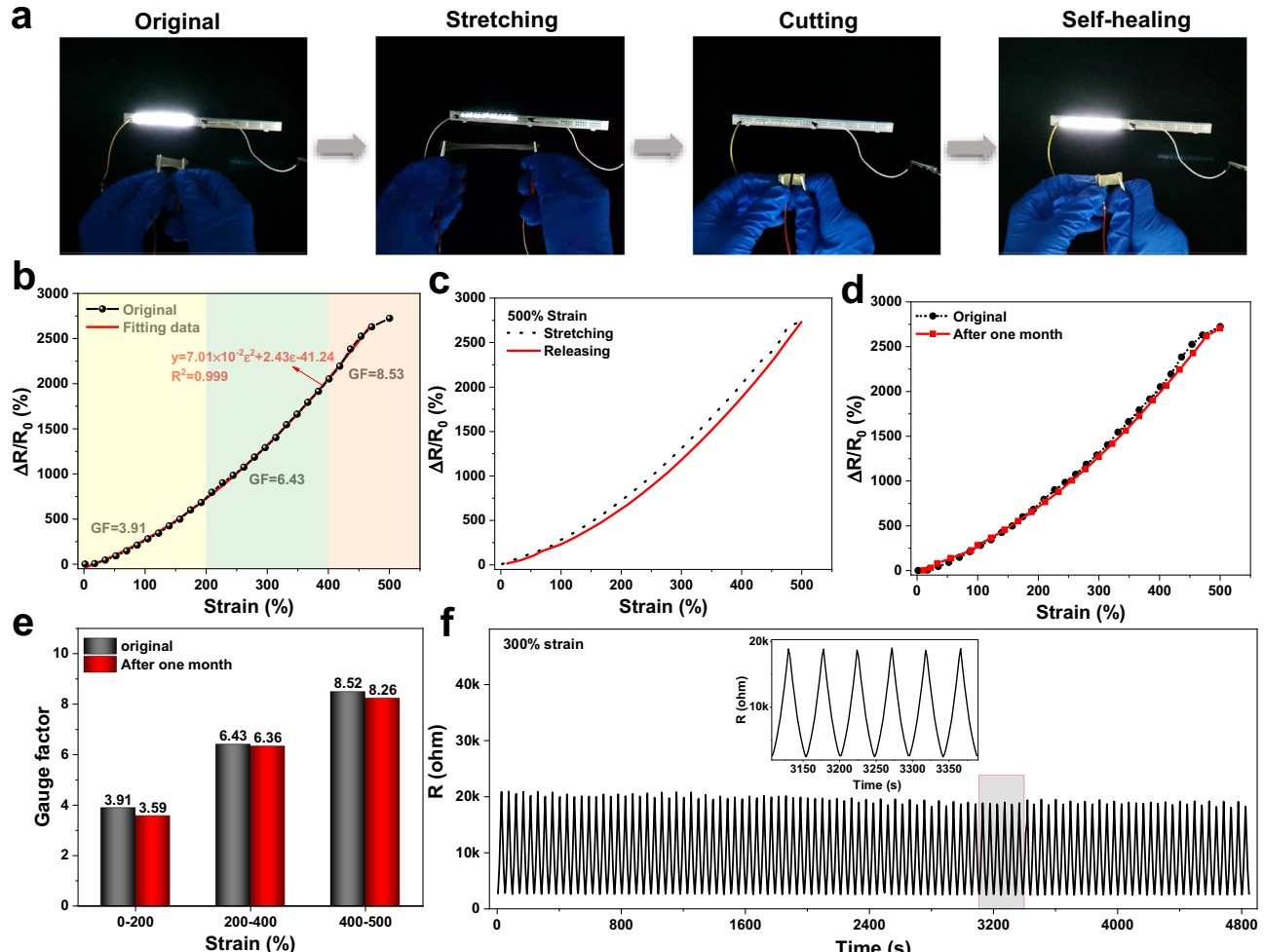

**Fig. 5 | Stretch-break-heal test and relative resistance changes. a** Stretch-break-heal test in a circuit of the PR-Gel (40 w/v%), confirming the highly conductive, stretchable and self-healing properties; **b** relative resistance changes of PR-Gel (40 w/v%) as a function of tensile strain; **c** relative resistance changes of PR-Gel (40 w/v%) in tensile loading–unloading curves at a strain of 500%; **d** Relative resistance changes of the original and one month placed PR-Gel (40 w/v%) sample as a function of tensile strain; **e** gauge factors of the original and one month-placed PR-Gel (40 w/v%) sample under various strains; **f** relative resistance response of PR-Gel (40 w/v%) under successive 100 loading–unloading cycles with a strain of 300%. Inset: enlarged view of six successive cycles.

returned to its original state. This illustrated that the PR-Gel-based strain sensor was flexible enough to detect the mobility of the finger with potential applications. Meanwhile, pressing also produced the variations in the resistance of the PR-Gel-based strain sensor and the variation rates of resistance remained consistent during the several repeated pressing cycles (Fig. 6b). From a subtle swallowing movement of the throat to wrist bending, and to large elbow flexion (Fig. 6c), all were captured by the PR-Gel based strain sensor. The variations in the electrical signal were completely synchronized with the changes in body movements without any lag, which indicated that this PR-Gel based strain sensor showed highly sensitive sensing performance and repeated stability.

**Real-time detection of the 3D printed sensor**
Traditional photocuring methods using slotted molds make it difficult to achieve more complex geometries and to accommodate structurally complex human bodies. 3D printing enables to fabricate reproducible and structurally complex devices, which has already revolutionized the manufacturing industry field[48–50]. The photopolymerizable crosslinker provides the suitability for digital light processing (DLP) based 3D printing to create customizable flexible sensors that combine the fatigue resistance and conductive properties of the PR-Gel. Lithium phenyl(2,4,6-trimethylbenzoyl)phosphinate initiated

rapid photopolymerization and tartrazine served as a light absorber to modulate the polymerization rate[51,52]. DLP based 3D printing transformed the PR-Gel precursor solution into solid 3D structures to fabricate PR-Gel based flexible sensors with high resolution and altitude complexity (Fig. 7a and Supplementary Fig. 22). The 3D printed PR-Gel based sensor could be stretched and maintained its intact structure with high resolution. The self-adhesiveness of the PR-Gel facilitated the direct attachment of the sensors to human skin without using glues. The 3D printed sensor was then attached to the hand back and deformed synchronously with movements such as finger pressing and joint bending (Fig. 7b–d) and showed a high sensitivity in variation rates of resistance due to the delicate 3D printed structure. After replacing the commercial electrode with 3D printed PR-Gel based electrode, stable and comparable real-time human electrocardiogram (ECG) signals were detected via an analog-to-digital converter and transmitted wirelessly to a computer, as shown in Fig. 7e. This proves that the designability of 3D printing can provide a new path to the development of PR-Gel based wearable flexible sensors.

## Discussion
A conductive PR-Gel system was designed and developed through the photopolymerization of a polymerizable pseudorotaxane crosslinker with acrylamide in the presence of choline chloride and in a binary

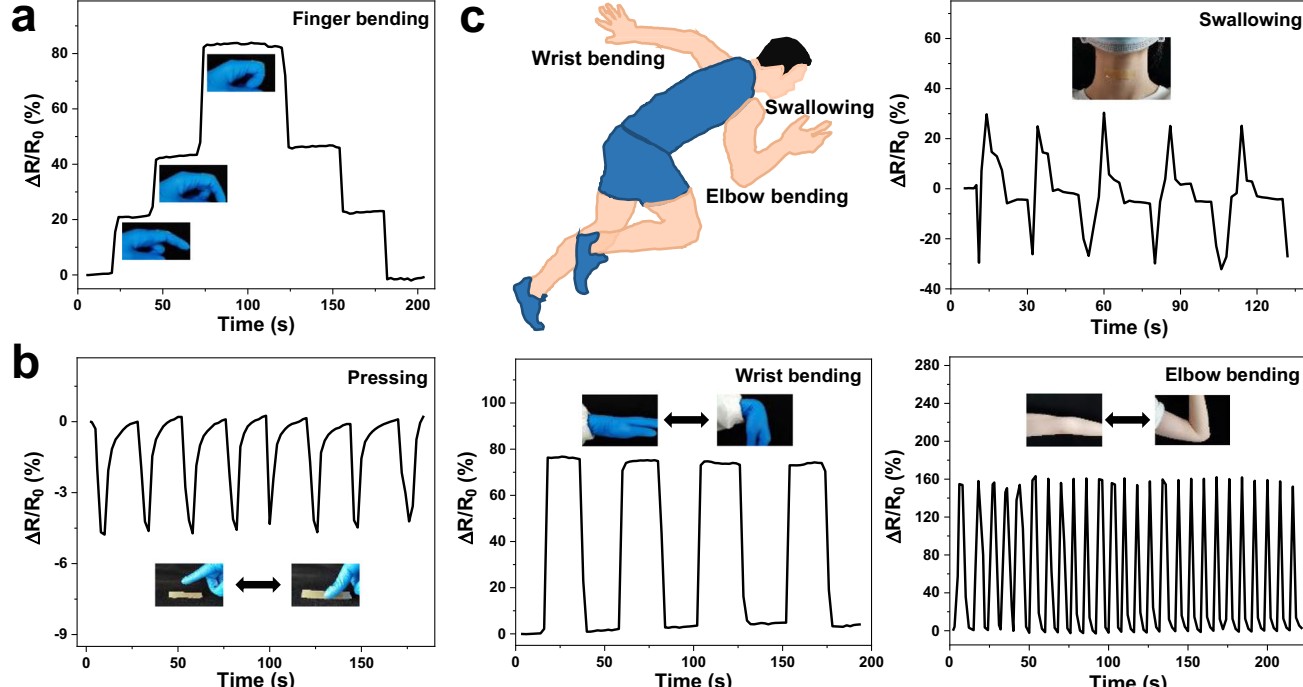

**Fig. 6 | Real-time human signal detection of the PR-Gel (40 w/v%) based flexible sensor. a** Monitoring of different finger bending angles in real-time by the PR-Gel-based flexible sensor through resistance variation; **b** relative resistance changes of the PR-Gel-based flexible sensor during pressing; **c** schematic illustration of the PR-Gel-based flexible sensor attaching to the throat, wrist, and elbow, and their real-time signal detections of human motions.

solvent system. The mobile junctions between β-cyclodextrin and bile acid units on the topological networks endowed the PR-Gel with extraordinary fatigue resistance, high strain sensitivity, fast self-adhesion and self-healing capability, and low temperature tolerance. PR-Gel based wearable sensors were successfully exploited to detect human motions in different modes, such as bending of fingers and wrist, as well as subtle muscle movements of swallowing, and exhibited excellent sensitivity, wide strain range and outstanding cyclic stability. The marriage of DLP 3D printing technology with PR-Gel realized personalized and rapid preparation of flexible strain sensors with complex geometry and high resolution. The 3D printed PR-Gel based electrode could detect human electrocardiogram signals in real-time with high repeating stability. This work not only represents one example of PR topological networks in the applications of hydrogel based wearable sensors, but also provides a new paradigm for the design of 3D printable flexible devices. Considering the precise and simple preparation, and excellent comprehensive performance, it is believed that this study offers new opportunities for the development of 3D printed smart and flexible electronic devices with PR topological networks in the future.

## Methods
### Materials
Acrylamide (Am, 99.9%), acryloyl chloride (96%), choline chloride (ChCl, 99%), β-cyclodextrin (β-CD, 98%), dichloromethane (DCM, 99.9%, with molecular sieves, water ≤ 50 ppm), 4-dimethylaminopyridine (DMAP, 99%), N,N-dimethylformamide (DMF, 99.8%, with molecular sieves, water ≤ 50 ppm), 4-(4,6-dimethoxy−1,3,5-triazin-2-yl)-4-methylmorpholinium chloride (DMTMM, 97%), ethylene glycol (EG, 98%), poly(ethylene glycol) (PEG, average Mn = 2000), rhodamine B, and tartrazine (yellow dye) were purchased from Macklin. 2-Hydroxy-4′-(2-hydroxyethoxy)-2-methylpropiophenone ($I_{2959}$, ≥98.0%), lithium phenyl(2,4,6-trimethylbenzoyl)phosphinate (LAP, ≥98.0%), lithocholic acid (LCA, >97.0%), N,N′-methylenebisacrylamide (MBA), sodium bicarbonate (NaHCO₃, ≥99.8%), and triethylamine (TEA, ≥99.5%) were

purchased from Aladdin. Acetone, diethyl ether and hydrochloric acid (HCl) were purchased from Guangzhou Chemical Reagent Factory (Guangdong, China).

### Synthesis of LCA-AC
A mixture of ethylene glycol lithocholate (LCA-EG, 420.6 mg), synthesized following methods in the literature[36], triethylamine (834 μL) and anhydrous dichloromethane (5 mL) was stirred well and then placed in an ice-water bath. Acryloyl chloride (325 μL) was slowly added dropwise under a N₂ atmosphere with stirring for 10 h. After removal of the precipitate by filtration, the filtrate was washed in saturated NaHCO₃ and deionized water, respectively. After evaporating the solvent and drying under vacuum, the obtained slightly yellow powder was noted as LCA-AC (435.7 mg, 85%). ¹H NMR (400 MHz, DMSO-d₆, ppm): δ 6.32 (2H, m, acrylate-*H*), 6.19 (2H, m, acrylate-*H*), 5.97 (2H, m, acrylate-*H*), 4.71 (2H, m, 25-C*H₂*), 4.30 (2H, m, 26-C*H₂*), 2.34 (2H, m, 23-C*H₂*), 0.92 (3H, s, 21-C*H₃*), 0.88 (3H, d, 19-C*H₃*), 0.66 (3H, s, 18-C*H₃*). ¹³C NMR (150 MHz, DMSO-d₆, ppm): δ 173.62, 165.78, 165.42, 132.41, 131.67, 129.25, 128.46, 74.34, 62.69, 62.16, 56.18, 55.93, 42.73, 41.64, 35.77, 35.19, 34.95, 34.65, 32.30, 31.01, 30.94, 28.14, 27.03, 26.71, 26.39, 24.27, 23.48, 20.88, 18.52, 12.31. HRMS-ESI Calculated for C32H48NaO6, [M + Na]⁺ 551.3349. Found: 551.3359.

### Synthesis of CD-AC
β-CD (2.0 g) was dissolved in an ice-water bath in anhydrous dimethylformamide (15 mL) and acryloyl chloride (1.67 mL) was added dropwise under N₂ atmosphere, which was stirred for 10 h. After removal of the precipitate by filtration, the filtrate was added dropwise to acetone, and then the precipitate was washed with acetone and dried in vacuum to obtain a white powder (2.23 g) of CD-AC in 90% yield. Acrylate units on each CD-AC are estimated to be ca. 1.9 on average. ¹H NMR (400 MHz, D₂O): δ 6.34 (1.9H, m, acrylate-*H*), 6.15 (1.9H, m, acrylate-*H*), 5.94 (1.9H, m, acrylate-*H*), 4.97 (7H, m, 1-*H*), 3.86 (7H, m, 3-*H*), 3.77 (16H, m, 5,6-*H*), 3.50 (14H, m, 2,4-*H*). ¹³C NMR (150 MHz, D₂O): δ 201.89, 132.85, 126.93,

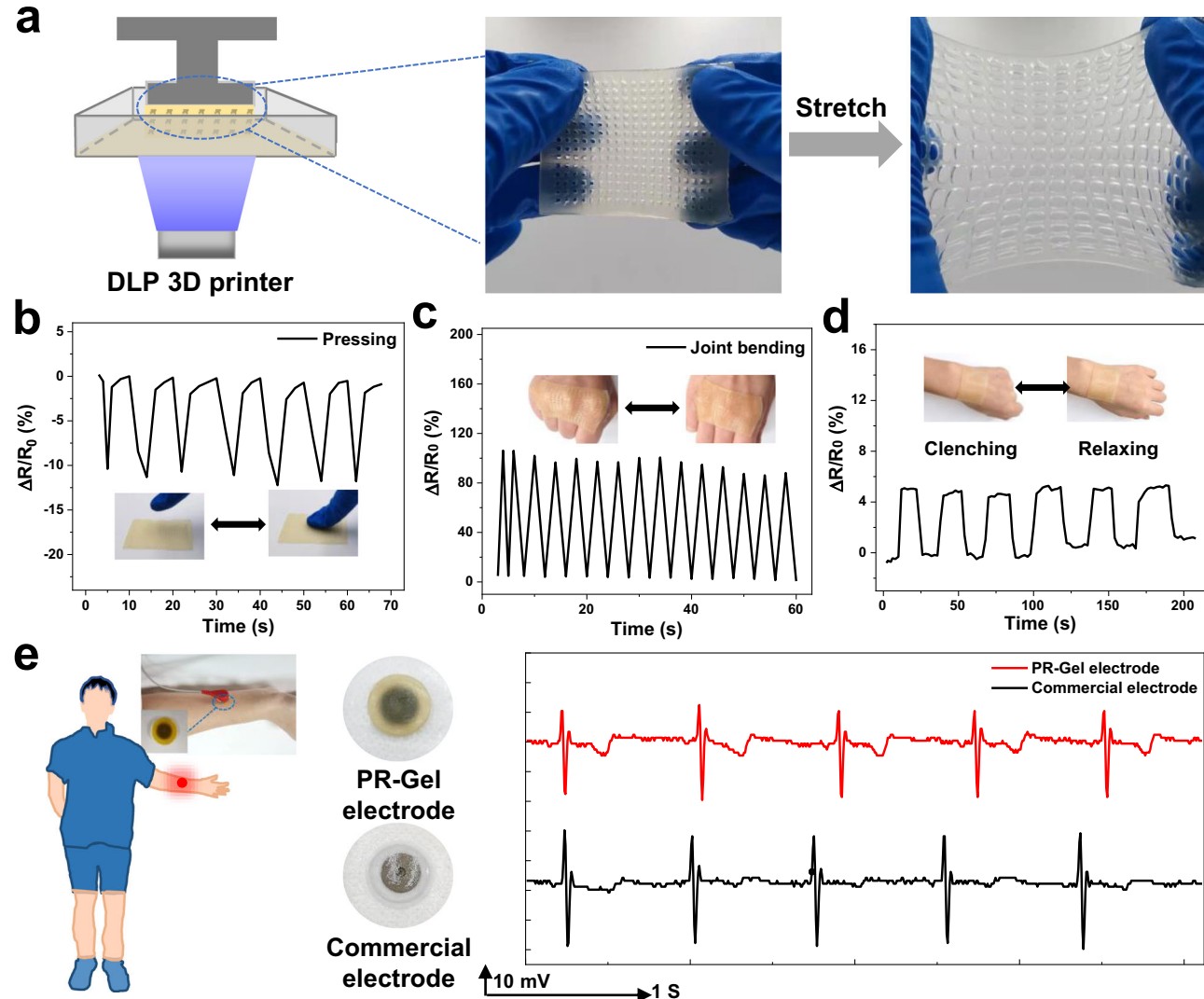

**Fig. 7 | 3D printed wearable sensors based on PR-Gel. a** Schematic illustration of the DLP based 3D printing configuration and 3D printed PR-Gel (40 w/v%) based wearable sensor with excellent stretchability, high resolution and altitude complexity. Relative resistance changes of 3D printed PR-Gel (40 w/v%) based wearable sensor during **b** pressing, **c** joint flexion, and **d** fist-clenching/relaxing; **e** applications of 3D printed PR-Gel (40 w/v%) based electrode for real-time detection of human ECG signal.

124.28, 101.90, 80.95, 72.96, 72.01, 71.75, 69.39, 63.92, 60.22. The most abundant component was determined to be diacrylated β-CD, HRMS-ESI Calculated for C48H75O37, [M + H]⁺ 1243.3987. Found: 1243.3949.

### Preparation of PR-Gel

Mixture of LCA-AC and CD-AC (1/1 mol.) was stirred in EG for 24 h and then water equal to the volume of EG was added. A predetermined amount of Am, and ChCl was dissolved in the above mixture. After purging with $N_2$ for 20 min, the initiator $I_{2959}$ was added to the monomer solution. The reaction mixture was transferred into a laboratory-made glass mold with a seal. The polymerization was initiated by UV irradiation (LED UV curing lamp, 45 W) at room temperature for 1 h to obtain hydrogels with different concentrations as shown in Table S1 in the supplementary information.

### Synthesis of LCA-AC-PEG

LCA (753.2 mg) and PEG 2000 (4.8 g) were dissolved in DCM (40 mL). DMAP (48.8 mg) and DMTMM (553.4 mg) were added, and then stirred at room temperature for 48 h. The reaction solution was added dropwise to the diethyl ether solution and the precipitate was dried in

vacuum to obtain the crude product of LCA-PEG. The final LCA-PEG was purified by dialysis against water (molecular weight cut-off (MWCO) 2000) and freeze-dried to give a white powder of LCA-PEG (3.86 g, yield 82%).

LCA-PEG (1.5 g) was dissolved in anhydrous DCM (10 mL), TEA (352.6 μL) was added, stirred well and then placed in an ice water bath at 0 °C. Acryloyl chloride (230.2 μL) was slowly added dropwise under a $N_2$ atmosphere and stirred for 10 h. After evaporation of the solvent, the crude product was dissolved in deionized water and dialyzed in a dialysis bag for three days (MWCO 2000). The purified solution was then freeze-dried, yielding a white floppy product poly(ethylene glycol) lithocholate derived diacrylate (LCA- AC-PEG) (1.41 g, yield 90%).

### Preparation of CD-Gel and MBA-Gel

CD-Gel and MBA-Gel were prepared by the same method and recipe as PR-Gel except for the crosslinkers as shown in Supplementary Table 1.

### Nuclear magnetic resonance (NMR) spectroscopy

¹H and ¹³C NMR spectra were acquired on a Bruker AVANCE 400 (400 MHz) spectrometer. 2D NOESY NMR spectra in D₂O were recorded on an AVANCE III HD 600 (600 MHz) spectrometer.

## High performance liquid chromatography (HPLC)

The CD-AC was determined with an Agilent 1260 high-performance liquid chromatography system (consisting of a G7111A Pump, G7115A DAD WR, G7129A vial sampler) equipped with a Diamonsil C18 column (150 mm by 4.6 mm, 5 μm; Agilent Technologies Inc. CA, USA). The mobile phase was acetonitrile delivered at a flow rate of 0.5 mL/min. The UV detection wavelength was 196 nm, the column temperature was 30 °C, and the injection volume was 10 μL.

## High resolution mass spectrometry (HRMS)

HRMS was performed on a Bruker maXis impact. The spectrometer parameters were set as follows: ESI (positive ion mode), capillary: 3500 V, end plate offset: −500 V, nebulizer: 0.6 Bar, dry heater: 180 °C, drying gas flow rate: 4.0 L/min. CD-AC was dissolved in methanol (100 μg/mL) and LCA-AC was dissolved in methanol/tetrahydrofuran (100 μg/mL, 9/1 vol.).

## Isothermal titrations (ITC)

ITC was performed on a MICROCAL PEAQ-ITC microcalorimeter equipped with a 280 μL sample cell and a 38 μL syringe. LCA (0.5 mM) was dissolved in EG/water and transferred into the cell, CD-AC (5 mM) was dissolved in EG/water, loaded into the syringe and injected into the cell in 19 aliquots (2 μL each) at 150 s intervals. LCA-AC-PEG (0.5 mM) was dissolved in water and transferred into the cell, and CD-AC (5 mM) was dissolved in water, loaded into the syringe and injected into the cell in 19 aliquots (2 μL each) at 150 s intervals. The cell temperature was set at 25 °C in all experiments.

## Powder X-ray diffraction (XRD)

The powder XRD patterns were taken by Cu Kalpha radiation with an Empyrean X-ray diffractometer. The powder sample was placed on the glass sheet recess and scanned in the 2θ range from 5° to 60° at a scanning speed of 1°/min.

## Mechanical tests

To measure the mechanical properties of the gels, tensile tests were performed on a universal tensile tester (JHY-5000, China). Uniaxial tensile was measured at a stretching speed of 100 mm/min and a gel sample thickness of 1 mm. Tensile tests at different temperatures were also carried out to investigate the mechanical properties. In the cyclic test, the gels were first loaded to different strains at 100 mm/min and then unloaded at the same rate. Successive cyclic tests without any resting time were also performed in a similar way. The dissipated energies were estimated by the area between loading–unloading curves and the resilience was calculated as the ratio of energy recovered in the unloading process to the work done to the sample in the loading process.

Compression measurements were performed on a universal material testing machine (Instron 5967) at a compression rate of 20 mm/min at room temperature. To ensure accurate testing and to eliminate interface effects, a small amount of silicone oil was applied to the top of the cylindrical gel that contacted the pressure transducer.

The puncture test was performed on a texture analyzer (CT3 4500, Brookfield) where the sheet gel (1.6 mm) was fixed between two metal plates. Needles with outer diamters of 0.1 and 0.5 mm, were placed vertically and moved downward onto the gel at a speed of 20 mm/min as shown in the videos in the Supplementary Information.

## Small angle X-ray scattering (SAXS)

SAXS experiments were performed at Xeuss 2.0 (Xenocs, France). The as-prepared 1 mm-thick hydrogel samples were cut into rectangles before they were analyzed. The samples were placed in a uniaxial stretching machine for deformation. The stretching direction of the hydrogel samples was vertical. The distance between the sample and the detector was approximately 2480 mm.

## Adhesion strength

The adhesion strengths of PR-Gel to various substrates were measured by lap-shear adhesion test. A piece of gel was sandwiched between two substrates and gently pressed. The test was then performed on a universal tensile tester (JHY-5000, China) at a speed of 100 mm/min. Adhesion strength was calculated by dividing the maximum load by the overlapping contact area.

## Rheological measurements

The rheological tests were performed on an Anton Paar MCR 302 air oscillating rheometer. The frequency scanning measurement was taken with a strain of 1% and a frequency range of 0.1–100 Hz. The self-recovery properties were analyzed by straining the hydrogels under an alternatively changing amplitude (1 and 500%) of oscillatory force at 1 Hz.

## Characterization of electrical properties

The ionic conductivity of the gel was measured by an electrochemical workstation (CHI 660D, CH Instruments, Inc. China). The ionic conductivity ($\sigma$, S/m) can be calculated by the following formula:

$$\sigma = \frac{L}{R \times A} \times 10^3 \tag{1}$$

where $L$ (mm) is the thickness of the gel, $R$ ($\Omega$) is the intersection of the Nyquist curve at the real part, which is the resistance of the PR-Gel, and $A$ (mm$^2$) is the effective contact area of the PR-Gel between the splints.

A electrochemical workstation was used to measure the relative resistance changes ($\Delta R/R_0$) at different strains. The gauge factor (GF) of the gel sensor can be calculated by the following formula:

$$GF = \frac{R - R/R_0}{\varepsilon} = \frac{\triangle R/R_0}{\varepsilon} \tag{2}$$

where $R_0$ and $R$ are the resistance of the gel before and after deformation, respectively, and $\varepsilon$ is the deformation strain. The prepared PR-Gel sensors were then attached to the body of the volunteer to measure $\Delta R/R_0$. Written informed consent was obtained from the volunteer before the measurements.

## SEM measurement

Three different gel samples were dialyzed for 3 days and then freeze-dried in a refrigerated drying chamber for 2 days. Then, the sample was sputter-coated with gold. A scanning electron microscope (SEM, Merlin, Zeiss, Germany) was used to observe the microstructure of the gel.

## Electrocardiogram (ECG) signal measurements

The PR-Gel electrodes for ECG signal measurements were prepared by replacing the adhesive layer on the commercial electrodes. Three prepared electrodes were then attached to the left and right arms and the right side of the body of the volunteer. Written informed consent was obtained from the volunteer before the measurements in this study. The signals were recorded and processed by an analog-to-digital converter and transmitted wirelessly to a computer via an adapter. The participant was author of this paper, and consent was obtained from research participant before conducting the experiments.

## X-ray photoelectron spectroscopy (XPS)

The powder sample of CD-AC was directly glued to the conductive adhesive for XPS testing on Escalab Xi+ (Thermo Fisher Scientific). The acquisition parameters were set as follows: pass energy 30.0 eV,

source gun type: Al K Alpha, Spot Size: 650 μm, energy step size: 0.100 eV.

## 3D printing of the PR-Gel

The precursor solutions used for printing PR-Gel are as follows. Mixture of LCA-AC and CD-AC (1/1 mol.) was stirred in EG for 24 h and then water equal to the volume of EG was added. A predetermined amount of Am, and ChCl was dissolved in the above mixture. After mixing uniformly, the initiator LAP (5 mg/mL) and tartrazine dye solution (0.1 mg/mL) were added to the monomer solution. The concentrations of the printing gels are the same as the PR-Gel (40 w/v%) in Table S1. The solution was then loaded into a digital light printer (Bio-Architects® DLP, Regenovo). The layer height was 50 μm and the printing time of each layer was set to 2.88 s. The light absorber (tartrazine) improves the printing resolution of PR-Gel and can be removed by UV illumination.

## Data availability

The data generated in this study are provided in the manuscript and the Supplementary Information. The raw data can be found from the Source Data file. Source data are provided with this paper.

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

## Acknowledgements

This work was supported by the National Natural Science Foundation of China (22075087 (Y.G.J.), 22072047 (Y.C.), 32022041 (X.S.), U22A20157 (X.S.), and U1801252 (L.W.)), the Science and Technology Program of Guangzhou City China (202007020002 (X.S.)), and the Natural Science Foundation of Guangdong Province China (2019A1515011129 (Y.G.J.)). We thank Prof. Hua Dong for the help with electromechanical measurements. We thank Prof. Taolin Sun and Prof. Caizhen Zhu for their helpful discussion in SAXS measurements. We also thank Ms. Weimin Xiao and Mr. Yichun Wu for the help with HRMS measurements.

## Author contributions

Y.Z., Y.G.J., X.S., and X.X. designed the study, conducted the analysis, and prepared the manuscript; X.X., Y.C., M.L. and C.L. synthesized hydrogels and conducted experiments; X.X., H.L., and G.W. conducted mechanical tests; H.L. mainly took charge of the self-healing experiment and drew the scheme; X.S., Z.W., and X.X. conducted 3D printing experiments; X.X., L.W., X.S., Y.G.J., and Y.Z. provided discussion and analyzed the experiment; X.X., X.S.,Y.G.J., and Y.Z. contributed to the manuscript writing; Y.Z., Y.G.J., and Y.C., L.W. contributed to the manuscript revision.

## Competing interests

The authors declare no competing interests.
