## [Peer Review File · Nature Communications]

Polymerizable rotaxane hydrogels for three-dimensional printing fabrication of wearable sensorsREVIEWER COMMENTS

Reviewer #1 (Remarks to the Author):

Suggestion: This work has some major problems, it needs to be revised in depth and then it may be considered for publication.

Problem 1: the claim of the slide-ring network is not correct and the materials synthesis has major problems.

As shown in scheme 1 and SI, the crosslinker was prepared with the host-guest interaction using CD-AC and LCA-AC, and they are both small molecules. After copolymerization with acrylamide, the CD has very limited sliding distance (over the length of LCA). Thus, this network could only be called a network with rotaxane crosslinkers. It is not a slide-ring gel.

The synthesis of the molecular component is ill described. There is not clear ¹H NMR and ¹³C NMR of each building block with clearly annotated proton and carbon signals in the SI. Their purity can not be confirmed.

The synthesis of diacrylate-CD is not correct. The authors added slightly more than two equivalents of acrylate for reaction. In the reaction, it will generate mono-, di-, tri-, and multi-substituted CD mixture. In addition, for di-substituted CDs, there are a mixture of AB, AC, and AD-substitution on each glucose ring, plus 2-, 3-, or 6- position substitution. The authors created a super complex mixture of CD derivatives.

The host-guest interactions were studied using water-soluble polymer LCA-AC-PEG and CD-AC in D₂O but the gels were prepared in EG and water (1 to 1 v/v). It would be necessary to verify that the binding affinities would not be undermined in the presence of EG.

The materials fabrication study is comprehensive and well characterized. The synthesized gels exhibited improved mechanical properties in terms of stretchability and fatigue resistance compared to the control groups and were employed to construct resistance strain sensors. The strain sensors were able to detect multiple human body motions and demonstrated their potential as wearable devices.

Other comments:

1. The term 'fracture energy' was incorrectly used and should be replaced with 'work of rupture'
2. The sizes of metal needles in Figure 2(e, f) should be noted for better illustration.
3. When synthesizing LCA-AC-PEG as described on page S2 in the supplementary information, the authors used DMTMM instead of LCA as described in Figure S3(a). Is this a typo? The authors should address that if so. Additionally, how did the authors realize the mono-functionalization of PEGs? It would be necessary to note the substitution ratio of LCA on functionalized PEGs.
4. There are many typos and inconsistencies in the manuscript, making the manuscript less readable. Just give a few examples: the axis labels in Figure 6(e) are missing. The ¹H NMR spectrum of LCA-AC-PEG was mentioned to be shown in Figure S6 but Figure S6 is not an NMR spectrum. The authors should do thorough proofreading before re-submission.

Reviewer #2 (Remarks to the Author):

Slide-ring hydrogel is an important topic in material science since the “pulley effect” endows the SR materials with unique mechanical performance. In this manuscript, the authors reported a conductive slide-ring hydrogels through the photopolymerization of acrylamide with pseudo-rotaxane crosslinker in the presence of electrolytic choline chloride. The topological networks enables the SR-gel with an excellent mechanical properties including well stretchability and fatigue resistance. Meanwhile, the SR-gel was applied to wearable sensors to detect human motions with an excellent sensitivity. The results of the manuscript presented here is substantial and convincing. The research provides a new pseudo-rotaxane crosslinker based SR materials and will draw attention to develop more SR systems. Hence, I recommend it to be published in Nature Communications after minor revision, and the following issues should be addressed.

1. The authors declared that the SR-gels have excellent self-healing ability. In my opinion, the self-healing ability usually comes from noncovalent bonds or dynamic covalent bonds in networks. So, I wonder what is origin of the self-healing ability of SR-gels? Furthermore, in Figure 3c, how about the mechanical performance of the self-healed sample? In Figure 3d, the rheological experiment may prove the well deformation recover ability of the SR-gel, not the self-healing property.

2. During the process for preparation of SR-gel, the host may slide out from the guest due to the pseudo-rotaxane structure. Though the LCA-AC and CD-AC were used equimolar amount, the systems will exist H-G complex, free host and guest. So, the association constant (K_a) of the LCA-AC and CD-AC is important to determinate the percentages of the H-G complex and it would be better to determinate the K_a .

3. The sentence of “.....first example of SR topological networks.....” may improper, because Ref 25 has already reported. The “.....sacrificial noncovalent bonds.....” in pages 3, line 11 is not match with the cited references.

Reviewer #3 (Remarks to the Author):

This paper reports a tough and conductive slide-ring (SR) hydrogel applicable to wearable sensors and electronic skins by crosslinking acrylated β -cyclodextrin (β -CyD) with bile acid and. The SR gel shows high stretchability with superior fatigue resistance and less hysteresis, self-healing in air, and high adhesion to human skins. Some high performances of the SR gels for wearable sensors are demonstrated. Since the paper is well written and of high novelty and interest, I think that it can be published in Nature Communications. However, it seems like the experimental evidence is not enough to confirm the slide-ring effect and some additional measurements may be needed.

1) If the illustration of the high mechanical performance of the SR gels in Fig 1b is right, β -CyD should release bile acid at high elongation ratio. I think that some additional experimental data are necessary to confirm that.

2) While the comparison between SR- and CD-gel in mechanical properties are presented, LCA-AC can be used as a crosslinker of polyacrylamide. LCA-gel should be synthesized to compare to the SR-gels as well.

3) The temperature dependence of the mechanical properties of the SR gels may be interesting because temperature can affect the inclusion complex formation of β -CyD and bile acid. Have such experiments been done already?

Responses to reviewers' comments:

Reviewer #1 (Remarks to the Author):

Problem 1: the claim of the slide-ring network is not correct and the materials synthesis has major problems.

As shown in scheme 1 and SI, the crosslinker was prepared with the host-guest interaction using CD-AC and LCA-AC, and they are both small molecules. After copolymerization with acrylamide, the CD has very limited sliding distance (over the length of LCA). Thus, this network could only be called a network with rotaxane crosslinkers. It is not a slide-ring gel.

Response: We thank the reviewer for reviewing our article carefully and giving us instructive comments, and we have revised the manuscript accordingly. The slide ring (SR) hydrogels with topological networks consisting of mechanical interlocked units were initially developed by Ito and coworkers using the polyrotaxane (PR) of poly(ethylene glycol) (PEG) and α -cyclodextrin (α -CD) (Figure R1a), in which the "pulley effect" created by the mobile junctions (crosslinks) could effectively disperse the stress. To date, detailed studies on SR topological supramolecular networks are mostly limited to the PR of PEG and α -CD. The most important feature of SR hydrogels is the mobile junctions of the topological networks, which can disperse stress and confer excellent mechanical properties on materials.

Figure R1. Illustration of SR topological networks constructed by (a) PR of PEG with α -CD and (b) polymerizable pseudorotaxane crosslinker of BA with β -CD.

In the present study, the polymerizable pseudorotaxane crosslinker of LCA-AC with CD-AC generated a great number of mobile junctions similar to those of SR hydrogels (Figure R1b). Therefore, the resulting topological hydrogels are also denoted as SR-Gel in our study. Furthermore, the "pulley effect" of BA and β -CD in the deformation of SR-Gel was also confirmed by the small angle X-ray scattering (SAXS) patterns of the stretched SR-Gel. Finally, the SR-Gel we created is not only a new kind of SR hydrogel, and mobile junction but also simultaneously endowed SR-Gel with important advancements in the applications of strain sensors, compared with the SR hydrogels constructed by PR. We added some comments to the introduction.

2. *The synthesis of the molecular component is ill described. There is not clear ^1H NMR and ^{13}C NMR*

of each building block with clearly annotated proton and carbon signals in the SI. Their purity can not be confirmed.

Response: Guest LCA-AC was prepared according to the previous literature (*Steroids* **2005**, *70*, 531–537). The ^1H NMR, ^{13}C NMR and high-resolution mass spectrum of the guest LCA-AC are provided below in Figures R2 to R4. The purity of LCA-AC was quantified by an internal standard method of ^1H NMR spectrum, and the purity was calculated to be higher than 98% (Figure R5). We added these data to supplementary information (Figures S1 to S4).

Figure R2. Synthesis route of LCA-AC and its ^1H NMR spectrum obtained in DMSO-d_6 as well as the assignments of all peaks.

Figure R3. ^{13}C NMR spectrum of LCA-AC obtained in DMSO-d_6 and the assignments of all peaks.

Acquisition Parameter

Source Type	ESI	Ion Polarity	Positive	Set Nebulizer	0.6 Bar
Focus	Active	Set Capillary	3500 V	Set Dry Heater	180 °C
Scan Begin	50 m/z	Set End Plate Offset	-500 V	Set Dry Gas	4.0 l/min
Scan End	1500 m/z	Set Charging Voltage	0 V	Set Divert Valve	Waste
		Set Corona	0 nA	Set APCI Heater	0 °C

#	m/z	I	I %
1	353.2661	121110	2.8
2	381.2973	81511	1.9
3	541.1202	108733	2.5
4	551.3359	4395613	100.0
5	552.3388	1365252	31.1
6	553.3407	248691	5.7
7	615.1392	174537	4.0
8	616.1395	102343	2.3
9	617.1374	79083	1.8
10	685.4351	82173	1.9
11	689.1578	211996	4.8
12	690.1584	134668	3.1
13	691.1561	109614	2.5
14	763.1766	161506	3.7
15	764.1772	119152	2.7
16	765.1755	95956	2.2
17	837.1950	128005	2.9
18	838.1961	103395	2.4
19	839.1941	87623	2.0
20	911.2139	106331	2.4
21	912.2145	91365	2.1
22	913.2131	83984	1.9
23	985.2321	88440	2.0
24	986.2328	82975	1.9
25	987.2319	79595	1.8
26	1059.2511	74525	1.7
27	1060.2521	73998	1.7
28	1061.2507	69812	1.6
29	1079.6783	84700	1.9
30	1213.7802	66590	1.5

Figure R4. ESI-HRMS spectrum of LCA-AC. Calculated for $C_{32}H_{48}NaO_6$, $[M+Na]^+$ 551.3349. Found: 551.3359.

Figure R5. ¹H NMR spectrum of LCA-AC (5.08 mg) mixed with internal standard 2,5-dihydroxybenzoic acid (5.03 mg) in DMSO-d₆ and the assignments of the related peaks. The purity of LCA-AC was quantified based on the integration ratio of peaks “d” to “18”, and the content of LCA-AC was calculated to be ca. 98.9%.

3. The synthesis of diacrylate-CD is not correct. The authors added slightly more than two equivalents of acrylate for reaction. In the reaction, it will generate mono-, di-, tri-, and multi-substituted CD mixture. In addition, for di-substituted CDs, there are a mixture of AB, AC, and AD-substitution on each glucose ring, plus 2-, 3-, or 6- position substitution. The authors created a super complex mixture of CD derivatives.

Response: We agree with the reviewer's opinion that CD-AC is a kind of mixture of substituted CD. We also stated that the acrylate units on each CD-AC were estimated to be ca. 1.9 based on the ¹H NMR spectrum of CD-AC shown in Figure R6 in our previous version. We added more captions in Scheme 1 to avoid misunderstanding.

The hydroxyl group on the 6-position of β-CD is much more active than that on the 2,3-position, so theoretically, the main reaction sites for esterification occurred on the 6-position of β-CD. HPLC result of CD-AC further confirms that this mixture contains only three components, 82.12%, 13.53% and 4.07%, respectively, instead of a super complex mixture of β-CD derivatives (Figure R8). The most abundant component was determined to be disubstituted β-CD by mass spectrometry (Figure R9). It should be noted that the numbers of acrylates on CD-AC do not affect the construction of the slide

ring structure of the resulting gels.

Figure R6. Synthesis route of CD-AC and its ^1H NMR spectrum in D_2O and the assignments of peaks. Acrylate units on each CD-AC were estimated to be ca. 1.91 based on the integration ratio of peaks a, b or b' to peak 1.

Figure R7. ^{13}C NMR spectrum of CD-AC in D_2O and the assignments of peaks.

Figure R8. The HPLC result of CD-AC further confirms that this mixture contains the three main components, 82.12%, 13.53% and 4.07%, respectively.

Figure R9. The ESI-HRMS spectrum of CD-AC indicates that the most abundant component is the diacrylated β -CD. Calculated for diacrylated β -CD, $C_{48}H_{75}O_{37}$, $[M+H]^+$ 1243.3987. Found: 1243.3949

4. The host-guest interactions were studied using water-soluble polymer LCA-AC-PEG and CD-AC in D_2O but the gels were prepared in EG and water (1 to 1 v/v). It would be necessary to verify that the binding affinities would not be undermined in the presence of EG.

Response: The extremely low solubility of LCA-AC in water limits the preparation of solution for the isothermal titration calorimetry (ITC). Finally, K_a of host-guest complex of LCA toward CD-AC was determined by ITC in a binary solvent system of EG/water. The K_a the value is $2.36 \pm 0.18 \times 10^4 M^{-1}$ (Figure R10), a value close to that of LCA derived polymer with β -CD in pure water (ca. $2.08 \times 10^4 M^{-1}$, *Angew. Chem. Int. Ed.* **2016**, 55, 11979–11983), indicating that the binding affinities of LCA with CD-AC would not be undermined in the presence of EG.

Figure R10. Apparent reaction heat in calorimetric titrations of EG/water solution of CD-AC (5.0 mM) injecting into EG/water solution of LCA (0.5 mM) at 25 °C and their typical isothermal titration calorimetry (ITC) fitting curves.

To further confirm the complexation of LCA-AC with CD-AC, powder X-ray diffraction (XRD) measurements were used to investigate the solid-state structures of the host-guest complex of LCA-AC and CD-AC (Figure R11), where the diffraction peaks (2θ) of LCA-AC are observed at 13.9 and 15.4, and two broad peaks (2θ) at 12.5 and 18.4 are assigned to the characteristic diffraction peaks of CD-AC. All the diffraction peaks of LCA-AC completely disappeared in the host-guest complex. In contrast, the physical mixture of LCA-AC and CD-AC exhibits both characteristic diffraction peaks of LCA-AC and CD-AC. This indicates that CD-AC exclusively recognizes the LCA unit of LCA-AC in the solid-state. We added these data to Figure 1 of the text.

Figure R11. Powder XRD patterns of CD-AC, LCA-AC, LCA-AC and the CD-AC physical mixture (1/1 mol.), and LCA-AC/CD-AC host-guest complex (1/1 mol.).

Other comments:

The materials fabrication study is comprehensive and well characterized. The synthesized gels exhibited improved mechanical properties in terms of stretchability and fatigue resistance compared to the control groups and were employed to construct resistance strain sensors. The strain sensors were able to detect multiple human body motions and demonstrated their potential as wearable devices.

1. The term ‘fracture energy’ was incorrectly used and should be replaced with ‘work of rupture’

Response: This was corrected. Thank you.

2. The sizes of metal needles in Figure 2(e, f) should be noted for better illustration.

Response: It was added. Thank you.

3. When synthesizing LCA-AC-PEG as described on page S2 in the supplementary information, the authors used DMTMM instead of LCA as described in Figure S3(a). Is this a typo? The authors should address that if so. Additionally, how did the authors realize the mono-functionalization of PEGs? It

would be necessary to note the substitution ratio of LCA on functionalized PEGs.

Response: The synthesis route of LCA-AC-PEG is shown below (Figure R12). The first step is the reaction between LCA and PEG 2000 under the coupling reagent 4-(4,6-dimethoxy-1,3,5-triazin-2-yl)-4-methylmorpholinium chloride (DMTMM), which includes both LCA and DMTMM. DMTMM was added in the synthesis scheme.

Figure R12. Synthesis route of LCA-AC-PEG.

Only the monofunctional compound, LCA-PEG, is used in the synthesis of LCA-AC-PEG. There is no concern about the difunctional compound of PEG with two LCAs. During the coupling reaction of PEG with LCA, the CH₂Cl₂ solution of reagents was slowly stirred at room temperature for 48 h. The abundance of PEG molecules surrounding the LCA and the low reactivity of PEG toward LCA to form ester bonds, both ensure the relatively high selectivity of LCA with PEG, so that the PEG chain is terminated by only one LCA to obtain the mono-functionalization of PEG. The purification was performed by precipitation and dialysis against water. The structure of mono-functionalized PEG is confirmed in the ¹H NMR spectra of LCA-PEG and LCA-AC-PEG in Figures R13 and R14.

Figure R13. ¹H NMR spectrum of LCA-PEG in DMSO-d₆, and the assignments of signals.

Figure R14. ¹H NMR spectrum of LCA-AC-PEG in DMSO-d₆, and the assignments of signals.

4. There are many typos and inconsistencies in the manuscript, making the manuscript less readable. Just give a few examples: the axis labels in Figure 6(e) are missing. The ¹H NMR spectrum of LCA-AC-PEG was mentioned to be shown in Figure S6 but Figure S6 is not an NMR spectrum. The authors should do proofreading before re-submission.

Response: We made a thorough correction. Thank you.

Reviewer #2 (Remarks to the Author):

Slide-ring hydrogel is an important topic in material science since the “pulley effect” endows the SR materials with unique mechanical performance. In this manuscript, the authors reported a conductive slide-ring hydrogels through the photopolymerization of acrylamide with pseudo-rotaxane crosslinker in the presence of electrolytic choline chloride. The topological networks enables the SR-gel with an excellent mechanical properties including well stretchability and fatigue resistance. Meanwhile, the SR-gel was applied to wearable sensors to detect human motions with an excellent sensitivity. The results of the manuscript presented here is substantial and convincing. The research provides a new pseudo-rotaxane crosslinker based SR materials and will draw attention to develop more SR systems. Hence, I recommend it to be published in Nature Communications after minor revision, and the following issues should be addressed.

Response: We thank the reviewer for the useful comments and suggestions, and we have revised the manuscript accordingly.

1. *The authors declared that the SR-gels have excellent self-healing ability. In my opinion, the self-healing ability usually comes from noncovalent bonds or dynamic covalent bonds in networks. So, I wonder what is origin of the self-healing ability of SR-gels? Furthermore, in Figure 3c, how about the mechanical performance of the self-healed sample? In Figure 3d, the rheological experiment may prove the well deformation recover ability of the SR-gel, not the self-healing property.*

Response: Since certain acrylates on LCA-AC were not photopolymerized, the host-guest dynamic complexation of uncapped LCA and CD moieties may form and play a determinant role in the rapid self-healing of SR-Gel (Figure R15), while the hydrogen bonds between polyacrylamide chains may also contribute to the rebuilding of the dynamic networks of hydrogels. Meanwhile, the polymer chains of SR-Gel could pass through the crosslinks to maintain an isotropic structure even under elongation as shown in the small angle X-ray scattering (SAXS) patterns, which enhances the spatial network fluidity of the hydrogel and is conducive to the improvement of the self-healing performance.

Figure R15. Schematic illustration of the crosslinked network structure of the SR-Gel. The host-guest

dynamic complexation of uncapped LCA and CD moieties is marked with red rectangles.

The healed gels in Figure 3c could withstand tensile deformation without fracture at ambient temperature, in which it was observed that the elongation ratio of the healed SR-Gel reached 400%, as shown in Figure R16. The cylindrical sample seems softer than the film sample in Figure 1d in the text. The self-healing performance of SR-Gel was further evaluated, and structural damage and recovery tests of SR-Gel were executed by applying continuous strain sweeps with the alternating low (1%) and high (500%) oscillatory excitations. Figure 3d shows the storage modulus (G') and loss modulus (G'') of SR-Gel under continuous strain sweeps, demonstrating that the SR-Gel had rapid self-healing properties after damage.

Figure R16. Stress-strain curve of the SR-Gel sample heated for 5 min at room temperature.

2. During the process for preparation of SR-gel, the host may slide out from the guest due to the pseudo-rotaxane structure. Though the LCA-AC and CD-AC were used equimolar amount, the systems will exist H-G complex, free host and guest. So, the association constant (K_a) of the LCA-AC and CD-AC is important to determinate the percentages of the H-G complex and it would be better to determinate the K_a .

Response: The extremely low solubility of LCA-AC in water limits the preparation of solution for the isothermal titration calorimetry (ITC). Finally, K_a of host-guest complex of LCA toward CD-AC was determined by ITC in a binary solvent system of EG/water. The K_a the value is $2.36 \pm 0.18 \times 10^4 \text{ M}^{-1}$ (Figure R10), a value close to that of LCA derived polymer with β -CD in pure water (ca. $2.08 \times 10^4 \text{ M}^{-1}$; *Angew. Chem. Int. Ed.* **2016**, 55, 11979–11983), indicating that the percentage of the host-guest complex between LCA and CD-AC is dominant in the presence of EG.

The K_a for the host-guest complex of LCA-AC-PEG with CD-AC in pure water is $6.09 \pm 0.60 \times 10^3 \text{ M}^{-1}$ based on the ITC results (Figure R17), which is slightly lower than that of in the presence of EG. These data were also added to the text.

Figure R17. Apparent reaction heat in calorimetric titrations of CD-AC aqueous solution (5.0 mM) injecting into LCA-AC-PEG aqueous solution (0.5 mM) at 25 °C and their typical isothermal titration calorimetry (ITC) fitting curves.

3. The sentence of “.....first example of SR topological networks.....” may improper, because Ref 25 has already reported. The “.....sacrificial noncovalent bonds.....” in pages 3, line 11 is not match with the cited references. correcting

Response: Corrected. Thank you.

Reviewer #3 (Remarks to the Author):

This paper reports a tough and conductive slide-ring (SR) hydrogel applicable to wearable sensors and electronic skins by crosslinking acrylated β -cyclodextrin (β -CyD) with bile acid and. The SR gel shows high stretchability with superior fatigue resistance and less hysteresis, self-healing in air, and high adhesion to human skins. Some high performances of the SR gels for wearable sensors are demonstrated. Since the paper is well written and of high novelty and interest, I think that it can be published in Nature Communications. However, it seems like the experimental evidence is not enough to confirm the slide-ring effect and some additional measurements may be needed.

Response: We really appreciate the reviewer for the useful comments and suggestions, and we have revised the manuscript accordingly.

1) If the illustration of the high mechanical performance of the SR gels in Fig 1b is right, β -CyD should release bile acid at high elongation ratio. I think that some additional experimental data are necessary to confirm that. ?

Response: To confirm the "pulley effect" of BA and β -CD in the deformation of SR-Gel, small angle X-ray scattering (SAXS) patterns of the stretched hydrogels were investigated, as shown in Figure R18. As clearly seen, the SAXS pattern of the SR-Gel before elongation is isotropic. At strain of 100%, 300%, and 500%, there is no anisotropy in the binary solvent system of EG/water, indicating that the SR-Gels are very homogeneous without a higher-order structure. These results were attributed to the polymer chains freely passing through the crosslinks acting like pulleys to maximize the entropy, maintaining an isotropic structure even under elongation. (*Macromolecules* **2006**, 39, 7386-7391). These data were added to Figure 1k.

Figure R18. SAXS images of the SR-Gel with different extension strains during the uniaxial stretching process.

2) While the comparison between SR- and CD-gel in mechanical properties are presented, LCA-AC can be used as a crosslinker of polyacrylamide. LCA-gel should be synthesized to compare to the SR-gels as well.

Response: The extremely low solubility of LCA-AC in water or even in the binary solvent system of EG/water limits the preparation of chemically crosslinked hydrogels. To increase the solubility of LCA-AC, the host-guest complex of LCA-AC with β -CD was prepared and then acted as a crosslinker. A very soft, sticky substance (Figure R19) was obtained and could not be tested as a gel sample, probably due to the low crosslinking density. The host-guest complexation of LCA-AC with β -CD was also confirmed by powder XRD results as shown in Figure R20.

Figure R19. Photograph of the polyacrylamide gel crosslinked with the host-guest complex of LCA-AC with β -CD.

Figure R20. Powder XRD patterns of the β -CD, LCA-AC, LCA-AC and β -CD physical mixture (1/1 mol.), LCA-AC/ β -CD inclusion complex (1/1 mol.) and polyacrylamide gel crosslinked with the host-guest complex of LCA-AC with β -CD, (1/1 mol.).

3) The temperature dependence of the mechanical properties of the SR gels may be interesting because temperature can affect the inclusion complex formation of β -CyD and bile acid. Have such experiments been done already?

Response: The results of the dynamic viscoelasticity measurements of SR-Gel are shown in Figure R21. The storage modulus (E') and loss modulus (E'') both decrease gradually with decreasing temperature from -30 to 40 °C. The ratio of E'' to E' (i.e., the $\tan \delta$ value) of the SR-Gel decreases gradually within temperature range from -40 to 0 °C, but begins to increase in the temperature range from 0 to 40 °C, probably indicating the “sliding out” of the host–guest crosslinks as the temperature increases. In other words, the added energy on SR-Gel was more extensively dissipated at the high temperature range, based on the dynamic viscoelasticity measurements.

Figure R21. Dynamic viscoelasticity measurements of SR-Gel on a dynamic thermomechanical analyzer (DMA Q800).

REVIEWER COMMENTS

Reviewer #1 (Remarks to the Author):

There are a number of problems that remained in the revised manuscript. Hence, the reviewer suggests for a revision.

1. Is the synthesized material a 'slide-ring' gel? This reviewer doesn't agree with the authors. The length of LCA is only 3-4 nm, and b-CD has a height of nearly 2 nm. This means b-CD can only travel 2nm distance of sliding. Compared to those slide-ring gels CD has 10-20 nm travel distance, the mobility of b-CD here is highly questionable.

The synthesized material has good performance, however, this reviewer can not attribute the enhance performance is attributed to the 'pully effect'.

As reviewer 2 pointed out, they must be a substantial amount of CD and LCA copolymerized in their free form, which gives self-healing performance to the product.

This reviewer suspects that the number of host-guest network is somehow comparable to the non-bind form. The material's performance enhancement is mainly attributed to the dynamic host-guest binding in the gel.

2. Evidence of the purity of the diacylated CD is not sufficient enough. As the authors responded in the letter, they carried out HPLC analysis on the sample. However, the first peak retention time is only 1.5 minute. This suggests that the HPCL condition was not well chosen and the authors should carry out a more detailed chemical composition analysis. The authors should also be aware that the chemical drawing of AD disubstituted b-CD is technically incorrect. They should draw it as a CD with two substitutions without specifying the substitution position.

Reviewer #2 (Remarks to the Author):

The author have answered my concerned issues, and revised manuscript accordingly. I recommend it to be accepted in Nature Communications.

Responses to Reviewers' Comments:

Reviewer #1 (Remarks to the Author):

There are a number of problems that remained in the revised manuscript. Hence, the reviewer suggests for a revision.

Response: We thank the reviewer for the additional comments and suggestions. We have carefully revised the manuscript accordingly.

1. Is the synthesized material a 'slide-ring' gel? This reviewer doesn't agree with the authors. The length of LCA is only 3-4 nm, and b-CD has a height of nearly 2 nm. This means b-CD can only travel 2nm distance of sliding. Compared to those slide-ring gels CD has 10-20 nm travel distance, the mobility of b-CD here is highly questionable.

The synthesized material has good performance, however, this reviewer can not attribute the enhance performance is attributed to the 'pully effect'.

As reviewer 2 pointed out, they must be a substantial amount of CD and LCA copolymerized in their free form, which gives self-healing performance to the product.

This reviewer suspects that the number of host-guest network is somehow comparable to the non-bind form. The material's performance enhancement is mainly attributed to the dynamic host-guest binding in the gel.

Response: We thank reviewer's comments. In order to distinguish the gel we prepared from the slide-ring gel, the gel in this work was named as PR-Gel due to the presence of the topological networks constructed by polymerizable rotaxane (PR). The polymerizable crosslinkers could generate a great number of mobile junctions in PR-Gel similar to those of slide-ring gel. The mobile junctions between β -CD and bile acid units simultaneously endow PR-Gel with important advancements in the applications of strain sensors.

	α CD	β CD	γ CD
No. of Glucose Units	6	7	8
Cavity Diameter (Å)	4.7	6.0	7.5
Height of Torus (Å)	7.9	7.9	7.9

Figure R1. Chemical structures, geometric dimensions, and physical properties of α , β , and γ -CD. (Reproduced from A. Harada et al, *Acc. Chem. Res.* 2014, 47, 2128–2140).

First of all, the height of β -CD cavity is about 0.79 nm (**Figure R1**), and the monomer LCA-AC is estimated to be ca. 5 times the length of β -CD cavity, up to ca. 4.0 nm due to the long

steroidal skeleton. Therefore, β -CD units could move along the axle of bile acid skeleton. In addition, β -CD may also move along the chains of polyacrylamide in some cases as shown in Scheme 1. In the present work, the host-guest complexation has been proven by 2D NOESY NMR in solution and by powder X-ray diffraction (XRD) in a gel state. Meanwhile, the mobility of β -CD units has also been verified by small angle X-ray scattering (SAXS), and the excellent mechanical properties of PR-Gel should be attributed to such topological structure.

Figure R2. Host-guest complex crosslinker of LCA-monoAC with CD-AC and the applications in the preparation of HG-Gel.

Secondly, we synthesized the monoacrylated LCA derivative (noted as **LCA-monoAC**), which formed the host-guest complex with CD-AC and then copolymerized with acrylamide to afford the HG-Gel as shown in **Figure R2**. The corresponding characterization spectra are shown in **Figure R3** and **R4**. HG-Gel did not contain the mobile junctions of rotaxanes, but form a great number of dynamic host-guest crosslinks. The property of HG-Gel is completely different from PR-Gel as shown in **Figure R5**. For example, **Figure R5a** confirmed that HG-Gel exhibited a lower tensile strength and elongation than that of PR-Gel at the same concentration of crosslinkers. **Figure R5b** shows the loading–unloading cycle tests of the HG-Gel under various strains. Compared with PR-Gel, the loading–unloading curve of HG-Gel at 200% and 300% strain exhibited the obvious hysteresis, demonstrating that the topological network of HG-Gel is different with that of PR-Gel. Meanwhile, the corresponding dissipation energy of HG-Gel is much higher than that of PR-Gel as shown in **Figure R5c**. More importantly, host-guest dynamic complexation of LCA-monoAC with CD moieties endows the HG-Gel with the self-healing behavior, which agreed well with previous host-guest dynamic hydrogels constructed by β -CD and bile acid moieties (Chem. Mater. 2015, 27, 387–393; Macromolecules 2017, 50, 9696–9701; Biomacromolecules 2018, 19, 626–632). Therefore, we can draw a conclusion that PR-Gel properties are not induced by the dynamic host-guest crosslinking.

Figure R3. Synthesis route of LCA-monoAC and its ^1H NMR spectrum obtained in DMSO-d_6 as well as the assignments of the related peaks.

Figure R4. ^{13}C NMR spectrum of LCA-monoAC obtained in DMSO-d_6 and the assignments of the related peaks.

Figure R5. Comparison of HG-Gel with PR-Gel. (a) Tensile stress–strain curves of HG-Gel, PR-Gel, MBA-Gel and CD-Gel with a concentration of Am 40 w/v% at 25 °C; (b) Tensile loading–unloading cycles of HG-Gel with a concentration of Am 40 w/v% at different strains and at 25 °C; (c) Dissipation energy of HG-Gel and PR-Gel (40 w/v%) in a cycle of tensile at 100%, 200% and 300% strain, respectively; (d) Self-healing behavior of the HG-Gel (40 w/v%) at 25 °C (colored with rhodamine for visualization).

At the same time, in our previous work (Angew. Chem. Int. Ed. 2018, 57, 9008–9012 shown in **Figure R6**), the “three-arm” host–guest supramolecules between β -CD and adamantane derivatives were copolymerized by a UV-initiated polymerization to form the dynamic crosslinks in the hydrogel (HGSMs). The host–guest interaction enabled the HGSMs to rapidly self-heal, but did not possess the tensile properties of fatigue resistance as PR-Gel, which also indicated that the excellent properties of PR-Gel should ascribe to the mobile junctions of rotaxanes in PR-Gel.

Figure R6. (a) Construction of the host–guest supramolecule (HGSM) hydrogel based on the host–guest complex between β -CD and Adamantane derivatives; (b) Mechanical compression curves of HGSM Gels at different concentrations; (c) Compression modulus of HGSM Gels. (d) Cyclic compression test curves of HGSM Gel (0.11 mmolL^{-1}); (e) Mechanical tensile curve of HGSM Gels. (Reproduced from X. Shi et al, *Angew. Chem. Int. Ed.* 2018, 57, 9008–9012)

Finally, there will be some dynamic host–guest complexation inside the PR-Gel, which has been drawn and indicated in the text as shown in Scheme 1. However, the concentration of free host and guest can be calculated according to the value of the binding constant (K_a) of complex between LCA and CD-AC in a binary solvent system of EG/water. The K_a value is $2.36 \pm 0.18 \times 10^4 \text{ M}^{-1}$ based on the result of ITC in **Figure R7**, in which the concentration of host–guest complex (rotaxane) is 100 times higher than that of free host and guest, indicating that

the free host and guest would not determine the properties of PR-Gel due to the limited ratio. For better clarity, we have also revised the manuscript title slightly.

Figure R7. Apparent reaction heat in calorimetric titrations of EG/water solution of CD-AC (5.0 mM) injecting into EG/water solution of LCA (0.5 mM) at 25 °C and their typical isothermal titration calorimetry (ITC) fitting curves.

2. Evidence of the purity of the diacrylated CD is not sufficient enough. As the authors responded in the letter, they carried out HPLC analysis on the sample. However, the first peak retention time is only 1.5 minute. This suggests that the HPCL condition was not well chosen and the authors should carry out a more detailed chemical composition analysis.

Response: We thank reviewer's comments. HPCL condition for the analysis of CD-AC was optimized, and the details were supplied in Supplementary Information. The new HPLC analysis of CD-AC was shown in **Figure R8** and diacrylated β -CD exhibited a monodisperse peak, further confirming that the purity of diacrylated β -CD is higher than 83%.

进样结果

#	名称	信号说明	RT (min)	峰面积 (mAU-s)	峰面积 %	峰高 (mAU)	峰高 %	含量	浓度	开始时间 (min)	结束时间 (min)
1		DAD1A, Sig=196, 4 Ref=off	3.859	3565.261	83.800	435.676	97.64			3.614	4.074
2		DAD1A, Sig=196, 4 Ref=off	4.604	689.208	16.200	10.508	2.36			4.074	7.226

Figure R8. HPLC result of CD-AC further confirming that the purity of diacrylated β -CD in mixture is higher than 83%.

The results of elemental analysis show that the contents of C, O and H in CD-AC are 46.10%, 47.78% and 6.12% respectively, which is closer to the calculated values for diacrylated β -CD. In addition, the X-ray photoelectron spectroscopy (XPS) and powder XRD analysis of CD-AC were also provided in **Figure R9** and **Figure R10** respectively, both of which confirmed the chemical composition of CD-AC. These results were also added in Supplementary Information.

Figure R9. (a) Wide scan XPS spectrum of CD-AC, and (b) C 1s and (c) O 1s XPS spectrum of CD-AC, which are consistent with the results in literature (**Nat. Commun.** 2022,13, 4181).

Figure R10. Powder XRD patterns of CD-AC and β -CD, indicating that CD-AC shows an amorphous state compared with β -CD.

The authors should also be aware that the chemical drawing of AD disubstituted b-CD is technically incorrect. They should draw it as a CD with two substitutions without specifying the substitution position.

Response: We thank reviewer's advice, and the drawing was corrected in Supplementary Information.

Reviewer #2 (Remarks to the Author):

The author have answered my concerned issues, and revised manuscript accordingly. I recommend it to be accepted in Nature Communications.

Response: We thank the reviewer for the recommendation of publication.

REVIEWERS' COMMENTS

Reviewer #1 (Remarks to the Author):

The authors have properly addressed this reviewer's comments. The manuscript is ready for publication.

Responses to reviewer's comments

Manuscript ID: NCOMMS-22-27414B

TITLE: Polymerizable rotaxane hydrogels for three-dimensional printing fabrication of wearable sensors

Reviewer #1 (Remarks to the Author):

The authors have properly addressed this reviewer's comments. The manuscript is ready for publication.

Response: We thank the reviewer for reading this article carefully and giving instructive comments.